# Seagrass Invertebrate Fisheries, Their Value Chains and the Role of LMMAs in Sustainability of the Coastal Communities—Case of Southern Mozambique

Sádia Chitará-Nhandimo [1,*] , Assucena Chissico [1] , Marlino Eugénio Mubai [2] , António de Sacramento Cabral [3] , Almeida Guissamulo [1,4] and Salomão Bandeira [1]

1 Department of Biological Sciences, Eduardo Mondlane University, Maputo 1100, Mozambique; assucenachissico@gmail.com (A.C.); almeida.guissamulo@hotmail.com (A.G.); salomao.bandeira4@gmail.com (S.B.)
2 Department of History, Eduardo Mondlane University, Maputo 1100, Mozambique; marlinomubai@gmail.com
3 Ocean Revolution Mozambique, Inhambane Town 1300, Mozambique; ttoneco@gmail.com
4 Museum of Natural History, Eduardo Mondlane University, Maputo 1100, Mozambique
* Correspondence: sadiachitara@gmail.com

**Abstract:** Invertebrate gleaning within seagrass meadows is a common activity across eastern African communities that depend on fisheries for their livelihoods. Based on a case study of two contrasting sites, Maputo Bay (MB) and Inhambane Bay (IB), this study documents, both qualitatively and quantitatively, the gleaning activity, its value chain and stakeholders, paying particular attention to the recently created Locally Managed Marine Areas (LMMAs) within IB, boasting creativity in seagrass invertebrate fishery management. Twenty-four common edible species were identified for MB, and 15 for IB; nearly all gleaners were women and children. Our estimates indicate that about 7.7 and 7.6 tons of invertebrates are collected in the peak catch weeks (spring low tides) in MB and IB, respectively. Resources are caught and sold at local markets, food fairs (for IB only), and restaurants, as well as for direct household consumption. One thousand one hundred and seventy two (1172) hectares of LMMAs (corresponding to nearly 0.05 of IB) of fisheries management, together with existing community and other stakeholder engagement and intervention on value chains, are at the center of tangible invertebrate fishery management.

**Keywords:** invertebrate fishery; Mollusca; value chain; LMMAs; stakeholders; seagrass management; Western Indian Ocean

## 1. Introduction

Seagrass meadows are among the most important habitats in the marine shallow water. They provide ecosystem services that have been globally acknowledged as having quite a high economic value, providing a critical contribution for the livelihoods and wellbeing of many coastal and islander communities [1–3]. Seagrass plays an important role in fishing productivity, enabling a wider range of ecosystem services [4,5], supporting numerous charismatic faunal species such as dugongs, turtles, and seahorses [6,7]. They also represent an important cultural asset to the coastal people whose lifestyle is intrinsically associated with seagrass' provision of food, recreation, and spiritual fulfillment [8,9].

In the Western Indian Ocean region, seagrass meadows are a source of food security for many communities [8]; thus, an important source of protein for rural coastal populations comes from harvesting invertebrates in the intertidal zone [6,10,11]. For example, in Mozambique, a great diversity such as pearl oysters (*Pinctada capensis*), clams and mussels (*Meretrix meretrix*, *Anadara antiquata*, *Modiolus auriculatus*), snails (*Volema pyrum*), and crabs (e.g., *Portunus* spp.) are caught for domestic consumption and for sale [12,13].

Invertebrate gleaning is commonly observed in the intertidal zone of tropical and subtropical countries [10,14,15]; however, information on gleaning catches is not often quantified and is rarely included in fishery statistics [14]. Studies suggest that invertebrate gleaning has probably existed for thousand years [10,16], ensuring the livelihoods of many coastal communities. However, population growth and, consequently, the increase in harvesters have put the sustainability of these fisheries and the quality of the habitat at risk [5,10,14]. In addition, these fisheries receive scant attention by the managing authorities.

In general, there is a paucity of research on this traditional activity, with a few highlights on: [17] who documented the diversity of fish in seagrass beds around Quirimba Island, northern Mozambique; [8] delved into interactions between humans and seagrasses in a rural tropical economy of the east coast of Zanzibar; [4] analyzed the human impact on invertebrate abundance, biomass, and community structure in seagrass meadows in southern Mozambique; and [18] who focused on the socio-ecological drivers and dynamics of seagrass-gleaning fisheries. While invertebrate harvesting is crucial to the livelihoods of coastal communities, harvesting practices are often harmful to seagrasses. Studies have proven that gleaning activity can negatively affect the growth and survival of seagrasses, leading to reduced densities of the meadows; carbon stocks in the sediment can also be lost due to intensive harvesting [19,20]. Fishing intensity can result in seagrass degradation, which in turn affects the volumes of catches. Therefore, it is crucial to develop strategies for the sustainable management of the gleaning activity.

Establishing locally managed marine areas (LMMAs) [21] is a bottom-up approach branded as community-led conservation management and conservation [22,23] and the promotion of sustainable resource use [24], resulting in tangible outcomes such as an increase in fishery biomasses [25], strengthening community representatives and their interplay with stakeholders and other interested actors [26]. Community-managed no-take zones (delimited areas where any fishing activity is prohibited) and fishing closures within LMMAs can either be permanent or temporarily [27] and may rely heavily on the collective participation of other actors (such as NGOs, government), given underlying issues of resource depletion, a lack of alternative livelihoods, population growth [22], poverty, and low education levels.

Keeping this background in mind, this study focuses on the seagrass invertebrate fisheries in Maputo and Inhambane Bays (southern Mozambique), with a particular focus on the harvested species, people involved, and value chain. It provides an overview on how LMMAs in Inhambane Bay contribute to the improvement of the sustainability of coastal communities linked to invertebrate fisheries.

## 2. Materials and Methods

### 2.1. The Study Areas

This study covers Maputo Bay (MB—128,000 ha) and Inhambane Bay (IB—25,000 ha), Western Indian Ocean (WIO), both in southern Mozambique (Figure 1) [12,28,29]. IB sits right by the Tropic of Capricorn and is tropical; MB is further south, being more sub-tropical (Figure 1). Both the MB and IB climate are characterized by a cold and dry season (from April to August) and hot and rainy season (September to March) with an average annual rainfall of 800 mm and 927 mm for MB and IB, respectively [30,31].

In MB, the water depth in most parts is less than 10 m but can reach 20 m (north of MB); in IB, the average depth is less than 5 m (due to extensive intertidal areas), reaching up 10 m in channels. Tides are semi-diurnal at both sites, and the water temperature ranges from 16 °C to 25.5 °C and 21 °C to 27 °C for MB and IB, respectively [28,32].

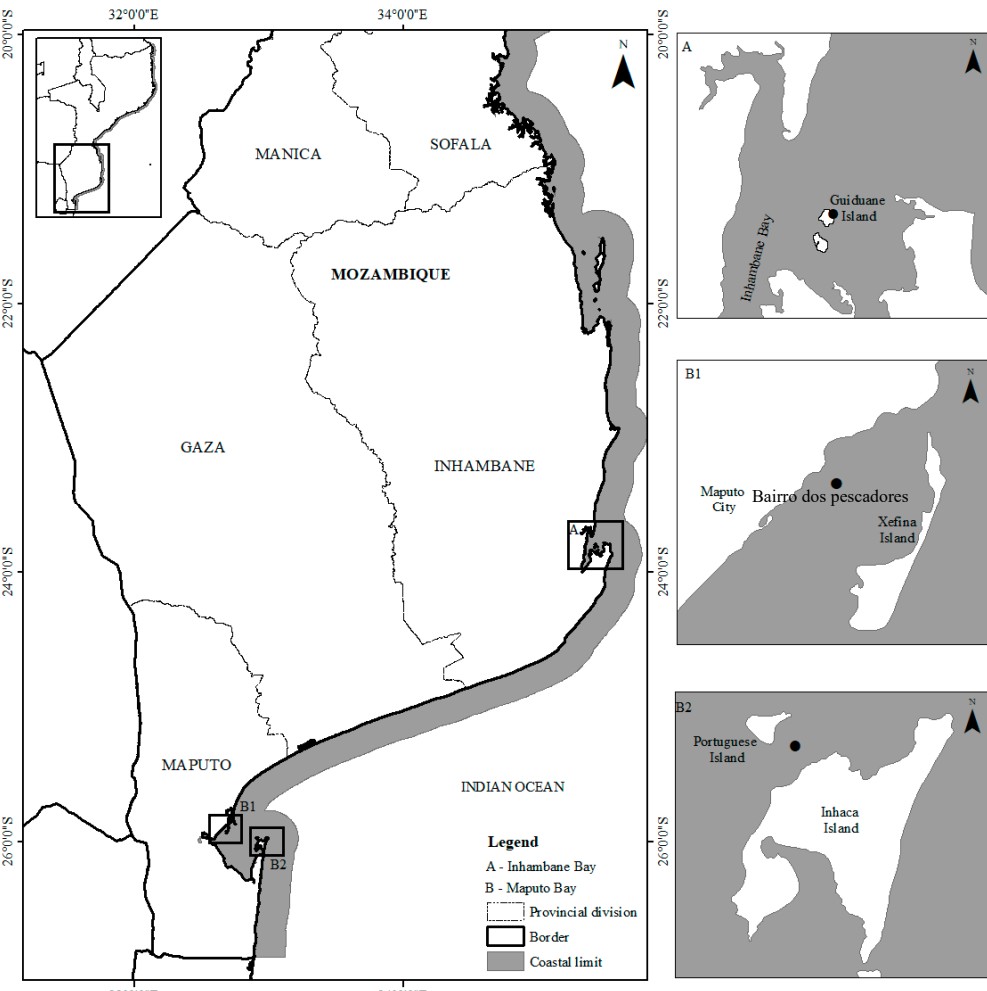

**Figure 1.** Study areas in southern Mozambique (map developed by J. Campira).

It is estimated that the population density in MB is around 3665 in habitants per km$^2$, mostly composed of young people aged between 20–24 years [12] and for Inhambane Province 21.8 inhabitants per km$^2$. Like in MB, the population is predominantly young [33].

The main habitats in MB are the following: mangrove forests (approx. 17,596 ha); submerged aquatic vegetation, essentially seagrass meadows and macroalgae with approx. 3875 ha; coral reefs and rock formations with around 54 ha [34]. IB has seagrasses meadows with 6199 ha [29] and 4000 ha of mangrove forests. Maputo Bay has 9 seagrass species while IB has 8. Seven (7) species, *Thalassia hemprichii* (Ehrenberg) Ascherson, *Halodule uninervis* (Forskål) Ascherson, *Thalassodendron ciliatum* (Forskål) den Hartog, *Oceana serrulata (R. Brown) Byng & Christenhusz* (former *Cymodocea serrulata*), *Cymodocea rotundata* Ehrenberg & Hempr. Ex Ascherson, *Halophila ovalis* (R.Br.) Hooker f., and *Syringodium isoetifolium* (Ascherson) Dandy, occur at both sites; *Zostera capensis* Setchellis is found only in MB; and *Enhalus acoroides* (Linnaeus f.) Royle in IB [29,35], with *Thalassodendron leptocaule* Maria C. Duarte, *Bandeira &* Romeiras found nearby hard subtract in rough waters, outside the secluded bays [36]. Table 1 below presents the seagrass area, its status and impacts, as well as risks and population densities around both bays.

**Table 1.** Brief summary of seagrass status and associated risks at the sites (adapted from [37]).

| Item | Site | |
|---|---|---|
| | Maputo Bay | Inhambane Bay |
| **Seagrass total area** | Inhaca Island 3943 ha<br>Bairro dos Pescadores 532 ha | 6199 ha |
| **Degraded seagrass area** | Inhaca Island 129 ha (3.2%)<br>Bairro dos Pescadores 459 ha | 5877 ha in 28 years (Between 1992 and 2020) |
| **Urbanization** | 1,914,130 inhabitants | Around 200,000 inhabitants |
| **Main contributing factor to seagrass degradation in order of importance** | Digging for invertebrate collection<br>Sedimentation due to flooding<br>Trampling | Several Cyclones |

*2.2. Sampling*

2.2.1. Assessment of Seagrass Invertebrates

Seagrass-meadow-associated invertebrates were assessed using standard protocols [38,39] as also summarized within the IPSN (Indo-pacific Seagrass Network) protocol (https://indopacificseagrass.network/research-protocols/, accessed on 9 January 2020). The IPSN protocol used essentially consists of: (1) the establishment of 50 m transects perpendicular to the coastline during low tide, separated by 10 m from each other; (2) the establishment of 0.5 m × 0.5 m quadrants every 5 m; (3) the collection invertebrate data in the quadrant (species identification and counting) [38–40].

MB data include samples made in 2017 and 2020, in three different seagrass communities predominated by *Oceana serrulata*, *Thalassia hemprichii*, and *Zostera capensis*; IB has samples from 2020, and the ecological surveys were made in *H. uninervis*, *T. hemprichii*, and *Cymodocea* spp. meadows. Eight transects were established in MB and twelve in IB; in each transect, ten quadrants where established. Within each of these quadrants, all the invertebrate species present were registered following identification guides for benthic invertebrates in tropical and subtropical areas [41,42]. Those species that were difficult to identify in the field were collected for further identification at the laboratory of Eduardo Mondlane University.

2.2.2. Interviews

Structured interviews with mainly open-ended questions [13] with invertebrate harvesters and sellers were carried out. In MB, interviews were carried out individually on the beach during the gleaning activity, coinciding with low tides, and in local markets. For IB, the interviews followed the pattern of household surveys with invertebrate gleaners, sellers, and key informants (local fishing leaders), and also in local markets. Household surveys were easily carried out in IB as most collectors live close to the sea, adjacent to the gleaning zone, with no obligation to limit the interview to the low-tide period.

The interviews aimed at evaluating mainly: (1) how invertebrate gleaning is carried out, which species are captured, and their purposes; (2) identifying the most important species and estimating the quantities of catches; (3) estimating perceptions of whether the quantities and the sizes captured have changed over the last 10 years; (4) finding the different stakeholders in the chain of capture, trade, and consumption of these invertebrates (sale forms and places, market research); (5) assessing the level of knowledge regarding the importance of seagrass for invertebrate fisheries; (6) determining whether local invertebrate fisheries management measures exist. Answers to the questions were filled in on previously prepared survey forms. The number of gleaners was counted during 10 days in Inhambane Island, and for more than a month in Bairro dos Pescadores, and then the average number of gleaners per day for both sites was calculated.

In MB, a total of 32 gleaners were interviewed from September 2020 to February 2021. In IB, a total of 39 gleaners were interviewed from November 2020 to May 2021.

In parallel to the interviews, we tried to estimate the duration of gleaning and the seagrass area that is disturbed during the gleaning activity. For this purpose, we monitored the gleaning activity in person with each gleaner: the activity was timed; we also visually analyzed how the gleaning instruments affect seagrasses; GPS was used to mark the starting location of the activity, the entire route taken by the gleaner to the end and then returned to the starting point. This monitoring was performedfor 20 days (1 harvester per day) in MB and 15 days (1 harvester per day) in IB. Using the software GIS-Arc View 10.3.1, the coordinates originated polygons that represent the space traveled by the gleaner during the invertebrate harvesting, and thus, we were able to calculate the disturbed area, assuming they collect all the way.

## 3. Results

### 3.1. Gleaners Profile

In the western Maputo Bay, Bairro dos Pescadores (BP) gleaners comprise residents of Bairro dos Pescadores (56%, $n = 18$), mostly adults between 20 and 40 years of age (62.5%, $n = 20$). In Guiduane island, Inhambane Bay, 95% of the gleaners are islanders aged between 30 and 60 years old (51.1%, $n = 18$); another well-represented group was the youngsters between 16 and 28 years old (21.3%, $n = 6$). Regarding education, 37.5% ($n = 12$) were illiterate and the majority (46.6%, $n = 13$) attended primary school in the BP; in Guiduane, the majority (71%, $n = 28$) attended primary school, 23.1%, ($n = 9$) were illiterate, and only a small minority attended secondary school (5.1%, $n = 2$).

The average number of gleaners per day is around 80 for MB and 40 for Inhambane Island. In MB, the interviewed gleaners were mainly women and young girls (93%, $n = 30$); men and young boys (7%, $n = 2$) can be found gleaning in two situations: young crab collectors or fishermen who for some reason did not go fishing in that day. On the other hand, in IB, gender roles in fishing are well defined, where men are engaged in fishing (fish with nets, line, boats) and women in gleaning; thus, all interviewed gleaners at Inhambane (Guiduane) Island were women (100%, $n = 39$).

On both sites, invertebrate gleaning is an income-earning activity for most gleaners. In MB, 25% of gleaners rely exclusively on the harvesting, processing, and sale of invertebrates for survival; 44% have gleaning as their main income activity; however, they have family members with other income-earning activities (farming, commercial activities), and 31% collect occasionally; this group includes maids, informal traders, and fishermen.

### 3.2. Gleaning Activity

Harvesting is carried out throughout the year in all extents of seagrass meadows at both sites, 2 weeks per month, during spring tide weeks (coinciding with a full moon and new moon), during low tide. The collection time per day lasts an average of 4 h and 30 min in the MB (BP), and around 3 h and 20 min in Inhambane Island.

Gleaning can be carried out manually, with bare hands in cases where the resource is easily observed and captured, such as in the case of gastropods, some clams, and crabs that remain on the surface of the sediment. Other invertebrates grow beneath the sediment and require the use of instruments to help with catching, such as spoons (for mussels), pliers (Prickly pen shell—*Pinna muricata*), and hoes and machetes for digging (razor clam) seen at BP. Crustaceans are also caught using fishing traps locally known as "Sengo" (rectangular baskets) and "gamboa"(reed walls that lead fish into narrow traps where they can be collected) placed along seagrass beds, a very common method in Inhambane Bay.

The invertebrate species captured do not vary much throughout the year, but the quantities do. According to the interviewees, larger amounts of bivalves and crustaceans are captured in the dry season (April to August); gastropods are captured in larger quantities during the rainy season (October to February). This pattern is usually also accompanied by the wandering around of gleaners from one specific seagrass area to another, as invertebrate species often occur in association with different seagrass species/communities.

### 3.3. Disturbance Area during the Gleaning Activity

Seagrass beds suffer some degree of disturbance during gleaning activity. This degree may vary according to the type of instrument used for harvesting. There are three main causes of disturbance at the sites: (1) the cut (lighter), as during the catching of crustaceans or gastropods with hands or pliers, pieces of leaves can be torn off; (2) trampling, an unavoidable but light disturbance on the seagrasses; and (3) excavation, very common in MB during the harvesting of *Solen cylindraceus* that lives buried in the sediment, causes the removal of vast areas of seagrass (Figure 2). Table 2 summarizes the main causes of disturbance and the average disturbance area per gleaner at the sites.

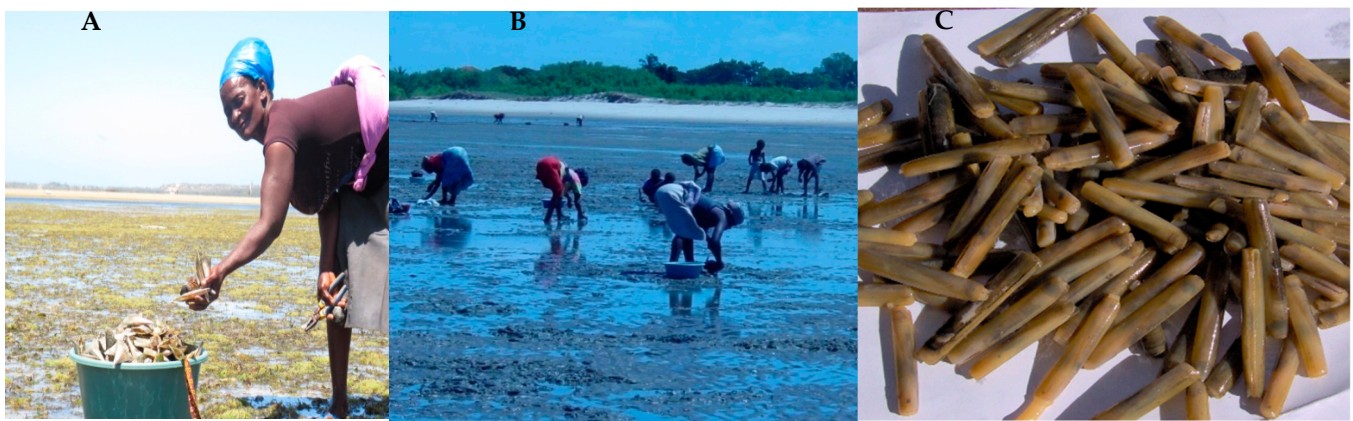

**Figure 2.** Harvesting *Pinna muricata* "mateo" with Pliers (IB) (**A**) and *Solen cylindraceus* "canivete" (western MB—(**B**,**C**)). Credits: Assucena Chissico (**A**), Salomão Bandeira (**B**,**C**).

**Table 2.** Causes of disturbance at the sites and estimate of average disturbed area per gleaner per day. Cr = *Cymodocea rotundata*; Os = *Oceana serrulata*; Ho = *Halophila ovalis*; Hu = *Halodule uninervis*; Tc = *Thalassodendron ciliatum*; Zc = *Zostera capensis*. Disturbance level: + lowly disturbed; ++ moderately disturbed; +++ severely disturbed.

| | Cause of Disturbance | Daily Disturbed Area/Gleaner | Seagrass Communities Affected |
|---|---|---|---|
| **MB** | Excavation, uprooting, trampling | 163.27 $\pm$ 26.98 m$^2$ | +++ Zc & Hu<br>+++ Zc, Ho& Hu<br>++ Ho & Hu<br>+ Ho |
| **IB** | Trampling, cut | 524.18 $\pm$ 2.07 m$^2$. | + Os & Hu<br>+ Os, Cr & Hu<br>+Os & Tc |

MB gleaners have a longer average time of permanence in the gleaning area; however, the disturbed area is smaller. This is because their harvesting methods by means of hoe, as well as machetes, especially in western Maputo Bay, are more labor intensive and require more physical effort. This aspect would be advantageous if the method was not highly harmful to seagrasses. In IB, the harvesting procedure is simpler, using less destructive tools, such as pliers and machetes, not requiring physical effort and therefore slightly disturbing a smaller area.

### 3.4. Harvested Species within Seagrass Meadows

Ecological surveys recorded over 89 species (grouped into 44 families) for both IB and MB, 65 species (41 families) in IB, and 28 species (19 families) in MB. Among these, 23 species in the western MB and only 11 within IB were edible species (Table 3). In MB, 10 bivalves, 8 gastropods, and 5 crustaceans were collected; and 6 bivalves, 2 gastropods, and 4 crustaceans were collected in IB (Table 4).

**Table 3.** Main invertebrate groups, families and number of species occurring at the sites.

| Group | Total Families | Main Families | Total Species |
|---|---|---|---|
| Bivalvia | 17 | Veneridae; Mactridae & Arcidae | 41 |
| Crustacea | 9 | Portunidae & Diogenidae | 19 |
| Gastropoda | 11 | Muricidae; Naticidae & Nassaridae | 20 |
| Echinodermata | 7 | Oreasteridae | 9 |
| **Total** | **44** | - | **89** |

**Table 4.** List of seagrass edible invertebrates collected in Maputo Bay (MB) and Inhambane Bay (IB).

| Groups | Families | MB | IB |
|---|---|---|---|
| Bivalvea | Veneridae | *Eumarcia paupercula*<br>*Meretrix meretrix*<br>*Tapes literatus*<br>*Tivela compressa*<br>*Vasticardium pectiniforme* | *Gafrarium pectinatum*<br>*Tapes literatus* |
| | Solenidae | *Solen cylindraceus* | |
| | Margaritidae | *Pinctada capensis* | *Pinctada capensis*<br>*Pinctada margaritifera* |
| | Mitilydae | *Perna perna*<br>*Modiolus philippinarum* | *Modiolus auriculatus* |
| | Tellinidae | *Serratina capsoides* | *Salmaco malitoralis* |
| | Pinnidae | | *Pinna muricata* |
| | Arcidae | *Barbatia decussata* | *Anadara antiquata* |
| Crustacea | Portunidae | *Portunus pelagicus*<br>*Portunus sanguinolentus*<br>*Scylla serrata* | *Portunus pelagicus*<br>*Portunus sanguinolentus*<br>*Scylla serrata* |
| | Callapidae | *Calappa hepatica* | |
| | Penaeus | | *Metapenaeus monoceros*<br>*Penaeus indicus* |
| Gastropoda | Melangenidae | *Volema pyrum*<br>*Volema paradisiaca* | *Volema pyrum* |
| | Muricidae | *Murex brevispina* | *Murex brevispina* |
| | Naticidae | *Polinices mammilla*<br>*Neverita didyma* | |
| | Conidae | *Conus tessulatus* | |
| | Cimatiidae | *Ranularia pyrum* | |
| | Strombidae | *Gibberulus gibberulus* | |

In brief, in MB, approximately 7.7 tons of edible invertebrates are collected per week during the low spring tide, which is when there is a peak in the gleaning activity and the greatest number of collectors, with an emphasis on razor clams (*Solen cylindraceus* about 2.9 ton) (see Table 5). At Guiduane Island (within IB), one week of spring tide allows the capture of a total of 7.6 tons of invertebrates, where mussels (dominated by *Modiolus auriculatus*) and prickly pen shell (*Pinna muricata*) contribute significantly (2.25 and 3.7 tons, respectively).

**Table 5.** Most commercially important resources and exploration details in Bairro dos Pescadores western Maputo Bay (MB) and Guiduane Island (IB).

| Site | Estimative on NoGleaners/Day | Groups | Species | Common Name | Local Name | BestExploitation Season | Daily Average Volume/ Gleaner (kg) | Estimate of the Weekly Average Volume/ Gleaner (kg) | Estimate of the Total Weekly Average Volume (in Metric ton) | Price/kg in USD |
|---|---|---|---|---|---|---|---|---|---|---|
| B.Pescadores—Maputo Bay (MP) | 44 | Gastropods | *Volema pyrum* | Pear melogena | Thokoma | Rainy season | 6.41 | 35.05 | 1.41 ton | 2.82 |
| | | | *Murex brevispina* | Short-spined murex | Xizenhe | | | | | |
| | | | *Polinices mammila* | Snail | Nforumana | Rainy season | 4.4 | 26.4 | 1.1 ton | 0.31 |
| | 65 | Bivalves | *Solen cylindraceus* | Razor clam | Nhengueta | Dry Season | 7.6 | 45.6 | 2.9 ton | 0.47–4.7 |
| | | | *Pinctada* sp. | Oyster | Mapalo | Dry Season | 9.69 | 48.45 | 1.45 ton | 0.15–0.23 |
| | | | *Anadara antiquata* | Pear melongena | Xilovo | Dry Season | | | | |
| | | | *Modiolus auriculatus* | Ear mussel | Mahoma | Dry Season | | | | |
| | 30 | Crustaceans | *Portunus pelagicus* | Blue swimming crab | Senze | Dry Season | 5.5 | 27.5 | 825 kg | 0.78–2.35 |
| | | | | Total | | | | | ~7.7 metric ton, weekly | |
| Island of Guiduane (IB) | 30 | Bivalves | *Pinna muricata* | Prickly pen shell | Matewo | Rainy season | ~15 kg fresh 1.7 (Processed) | ~75 kg fresh 8.5 processed | ~2.25 ton fresh | 1.25 |
| | | | *Pinctada capensis* | Oyster | Mapalo | Dry Season | ~10 kg fresh 1.0 kg processed | ~50 kg fresh 8 kg processed | ~1.2 ton fresh | 0.78–1.88 |
| | | | *Modiolus auriculatus* | Mussel | Mahoma | Dry Season | ~25 kg fresh 1.7 (Processed) | 125 kg fresh 8.5 processed | ~3.7 ton fresh | 0.78–1.88 |
| | 24 | Gastropods | *Volema pyrum* | Snail | Thokoma | Rainy season | | | | |
| | 15 | Crustaceans | *Portunus pelagicus* | Crab | Senze | Dry season | 5 kg | 35 kg | 0.52 ton | 0.47 |
| | | | | Total | | | | | 7.67 metric ton, weekly | |

## 3.5. Seagrass Resources Value Chains

The value chain comprises four components: harvesting, processing, commercialization, and consumption (Table 6). The first has been explained above. Processing is the treatment that resources go through for conservation without the risk of deterioration. Clams, razor clams, and crabs are usually sold fresh on both sites. Freezing and refrigeration with ice are the conservation methods mainly applied within MB fish markets, where resources have a relatively long waiting time before being sold. Two types of general processing are applied for gleaning in the study sites:

**Table 6.** Value chain components at each site, and stakeholders.

| | Bairro dos Pescadores | | Inhambane Island | |
|---|---|---|---|---|
| Component | Stakeholders/ Actors | Place of Origin/Place of Execution | Stakeholders/ Actors | Place of Origin/Place of Execution |
| Gleaning | Gleaners | Different districts of Maputo, but mainly from Bairro dos Pescadores | Gleaners | Inhambane Island Barra |
| Processing | Gleaners Sellers Restaurants | Different districts of Maputo, but mainly from Bairro dos pescadores | Gleaners Restaurants | Inhambane Island Barra, Tofo, and Inhambane City |
| Marketing | Gleaners Sellers Restaurants | Bairro dos Pescadores Market and Fishing Market at Costa do sol (Near BP) Markets in other districts in Maputo City | Gleaners Sellers Restaurants | Seafood trade fairs in Maxixe and Guiúa. Markets in other Inhambane districts; Markets at Maputo and Beira cities. Barra, Tofo and Inhambane City |
| Consumption | Consumers | Mainly form Maputo City | Consumers | Inhambane city and Maxixe |

Cooking and drying: invertebrates are boiled, removed from the shell, and then laid out to dry under the sun. This method is applied for gastropods in MB, and gastropods, mussels, and prickly pen shells in IB. Processing for sale is almost mandatory for all resources as they are not sold on the same day.

Smoking: this method is applied on shrimps in IB. Shrimps are sun-roasted in zinc sheets placed inside drums covered with cloth, and with coconut shell firewood underneath.

Commercialization and Consumption: in MB, the percentage of gleaners that capture for direct consumption is 18.8% ($n = 6$), while 53.1% ($n = 17$) commercialize these resources informally: at home, and at strategic points along the coastal districts; 47% ($n = 9$) are divided into six markets, four generalist markets in the interior of Maputo city (26%, $n = 4$), and two fish markets (21.7%, $n = 5$). Shellfish best appreciated fresh and therefore the conservation methods applied by gleaners and sellers in MB (mainly refrigeration) allow them to sell at moderately advantageous prices. IB has two well-structured and organized trade fairs, especially for the commercialization of these fisheries: the fair in Maxixe District (western IB), which takes place every Wednesday, and the fair in Guiúa (at the entry of Inhambane City), which runs on Fridays. By taking place on fixed days, these trade/gastronomy fairs are able to bring together wholesalers, retailers, and local and other district consumers at once, creating competitive prices and ensuring that the gleaner and primary seller are able to sell all products, without leftovers.

Figures 3 and 4 depict the value chain for both MB and IB. They show that a large part of the captured invertebrate fish goes to local consumers directly from the gleaners through local fish markets and restaurants (MB: Figure 3) or special fairs for shellfish fisheries (IB: Figure 4). In Inhambane, an established tourism hotspot in southern Africa, gleaners have mentioned having a direct link with restaurant owners who pay them to catch the resources. In both places, resources receive the highest prices from distant restaurants and markets. The IB resource market sphere has spread out to neighboring regions, also beyond Inhambane provinces, to places such as Beira (Further north) and Maputo.

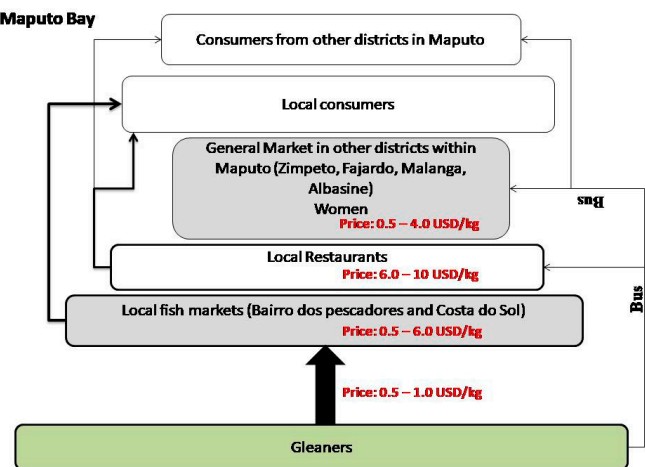

**Figure 3.** Illustration of the value chain: stakeholders, product flow, and market price variation between stakeholders for Bairro dos Pescadores (western MB) resources. Line thickness indicates the volume of resources transacted (thicker lines indicate higher volumes; thinner lines indicate lower volumes). Gleaners use buses to arrive at the marketplaces.

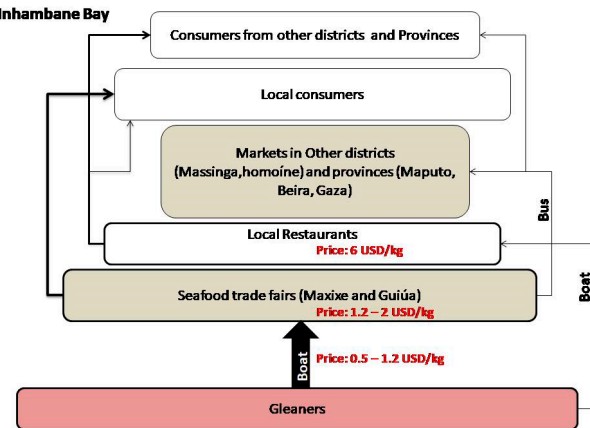

**Figure 4.** Illustration of the value chain: stakeholders, product flow, and market price variation between stakeholders for Inhambane Island resources. Line thickness indicates the volume of resources transacted (thicker lines indicate higher volumes; thinner lines indicate lower volumes). Gleaners use boats to arrive at the marketplaces.

An estimate for the monthly total revenue for gleaners in western MB is about USD 2630, mostly from razor clams, *Solen cylindraceus*. For IB, we estimated a USD 1229.7 monthly total revenue for gleaners, mostly from *Pinna muricata* and the pearl oyster *Pinctata capensis*. Such revenues would go higher with other stakeholders due to added value and tourism/restaurant business. Throughout the year, the value gained by gleaners can fluctuate over the months as resources peak at different times. For example, clams and oysters are abundant in the dry season, and gastropods in the rainy.

### 3.6. What Next for Gleaning and Teaming with LMMAs?

Resource degradation, both in quantity as well as in size, has been observed within the two sites. Compared to the last 10 years (from 2010 to 2020), it was observed that 40.6% (*n* = 13) of gleaners in the western MB claimed that the invertebrate amounts captured have decreased, while 53.1% (*n* = 17) do not know. A total of 92.3% (*n* = 36) of respondents in Guiduane/Inhambane Island within IB stated that the volume of catches has been degreasing, and 79.5% (*n* = 31) said that the sizes of bivalves have also significantly shrunk.

When asked about the reasons for these changes for BP (western MB), the majority of gleaners (50%, *n* = 16) said they do not know, 31% (*n* = 10) said it was due to changes in

climate (reduction in rainfall, higher temperatures), and 6.3% (*n* = 2) related to a reduction in seagrass areas. Gleaners at Inhambane island within IB listed several possible causes for the reduction in catches: (1) the growing number of gleaners (41.1%, *n* = 16); (2) lack of rain (28.2%, *n* = 11); the appearance of too many non-resident gleaners (10.3%, *n* = 4); the reduction in seagrass areas and increased capture of juveniles' (5.2%, *n* = 2); 20.6% (*n* = 8) of respondents do not know.

Faced with these constraints, we sought to find out from collectors what measures could be applied to improve the conditions of the activity, increase catches, and improve the quality of the resources. Within Bairro dos Pescadores (MB), all respondents answered that they did not know; in Guiduane (IB), 43.7% (*n* = 17) suggested the temporary interruption of gleaning (without mentioning periods) so that the resource can rest and recover; 12.9% (*n* = 5) suggested prohibiting the gleaning by non-resident collectors to reduce pressure, as well as collection through diving, which they said is harmful; 20.5% (*n* = 8) answered that they do not know. An analysis of issues on sustainable seagrass harvesting and mitigation measures are detailed in Table 7.

**Table 7.** Assessment of risk factors to gleaning sustainability on the sites, mitigation measures applied and suggested by gleaners.

| Site | Issues | Causes | Mitigation Measures Suggested by the Gleaners |
|---|---|---|---|
| **Bairro dos Pescadores/Maputo Bay** | General decline in invertebrate catches | Excessive number of gleaners | None |
| | | Decline in the seagrass area | None |
| **Inhambane Island** | General decline in invertebrate catches and quality (size) | Excessive number of gleaners | Establishment of harvesting closure periods; |
| | | Early resource catch (collection of juveniles) | Establishment and dissemination of minimum catch sizes |
| | | Decline in the seagrass area | Restoration of degraded seagrass areas |
| | | Territorial conflicts | Recollection ban for non-residents |

### 3.6.1. Policy Framework

Within Mozambique, five instruments were assessed as being the main ones concurring in the protection of marine and coastal habitats such as the seagrass meadows: The Fisheries Act 22/2013; The Framework Environmental Act 20/97; The Environmental Impact Assessment Process 54/2015; Law of the Sea no. 21/96; The Land Law no 19/97; and the new Maritime Fisheries Regulation (Decree 89/2020).

The Fisheries Act 22/2013 also deals with General Regulation for Maritime Fisheries (89/2020), standing for the conservation and proper use of aquatic biological resources and their respective ecosystems. It allows the establishment of marine protected areas for fishing resources and provides parameters for the conservation, preservation, and management of fishery resources, considering species and fishing areas, as well as the need for the protection of marine mammals and other rare or endangered species (e.g., dugongs and seahorses). This fisheries instrument regulates fishing practices such as the type of gear, fishing licenses, and engagement for the fishing closure season. However, it does not mention gleaning activities and the management of this type of activity widely carried out by the most vulnerable people, such as woman and children.

The Framework Environmental Act No. 20/97 establishes the general regime for the protection of biodiversity, discouraging practices that do not respect the environment, and acts in favour of the conservation of biological resources such as seagrass and threatened species dependent on them, such as dugongs. Furthermore, this national instrument gives authority to the government to strengthen and ensure the implementation of needed

management of coastal and marine habitats and promotes the existence of a conducive environment for the regeneration and restoration of animal species and recovery of habitats.

The Environmental Impact Assessment (EIA) Process 54/2015 regulates the environmental transformation or impact promoted by enterprises. It is also a sensitive and very relevant instrument to the coastal and marine habitats considering its role in regulating activities that may impact the environment, including seagrass habitats, as a result of planned development projects. The EIA instrument relies on several standards based on the impact. It also incorporates high-impact activities such as ports, oil, gas, and others, demanding compensation and offset mechanisms.

The Land Law no 19/97 states that the first 100 m from the high tide mark to inland is a public domain. This law defines areas of total and partial protection, and shallow-water habitats such as mangroves and seagrasses are included in the latter category. This area is exempted from the allocation of land-use rights (DUATs).

The new Maritime Fisheries Regulation (REPMAR), Decree 89/ 2020 defines, as a measure of fisheries management, the limitation of the fishing activity (fencing or closing of fisheries), and limitation of the catches volume. This legal document also supports the promotion of the participation of fishing communities in the planning and application of fishery management measures.

### 3.6.2. Establishment of LMMA's and Role of Key Actors in Community Engagement

The establishment of locally managed marine areas (LMMA's) in Inhambane Bay coupled with recent approval of the new Fisheries Law no. 22/2013 and Maritime Fisheries General Regulation (Decree 89/2020) offers a new platform for sustainable use and conservation of marine resources, specially in Inhambane Bay, where tangible actors appear to have devoted comparatively more time and resources to community management of shallow-water habitats and resources. LMMAs in IB cover 1172 ha, all proclaimed in 2017.The procedures for the development and proclamation of these areas were bottom up, involving first having discussions and agreements within communities, then follow-up discussionsat the district level (surrounding the IB), then at the provincial, then national level. The process involved the Districts of Inhambane, Maxixe, Morrumbene; the municipality of Inhambane City, the local governments of the district of Jangamo and the traditional Chiefs, elected leaders and community fishing councils (CCPs) of the Villages of Muele, Nhampossa, Mucucune, Guiduane, Marrambone, Maxixe, Morrumbene, Chamane, Chicuque, Nguja, Kuguana and Madava. The Ministry of Fisheries has designated 11 areas under National Fisheries Law 22/2013 and one under the Regulation 89/2020. The Community Fishing Councils (CCPs) manage this network of LMMAs under the authority of the local, district, and provincial governments and Ministry of Sea, Inland Waters and Fisheries, with the cooperation of the Marine and Coastal Police (PRM), and the NGO Ocean Revolution Mozambique. Nearly 5% of Inhambane Bay comprised the 12 LMMAs detailed in Table 8 below.

The main goals of the establishment of LMMAs in Inhambane Bay (Figure 5) include the conservation of biodiversity and the recovery and growth of the seagrass habitats to support the sustainable use of marine resources in the entire bay. Within these goals is the management plan for these areas foresees, in addition to other aspects related to the various habitats of the bay, to sensitize and educate the communities about the ecological importance of seagrass for the sustainability of the IB fisheries activity. Within these LMMAs, the practice of fishing, tourism, rituals, and construction is expressly prohibited. Local communities within Inhambane Bay, represented by the community fishing councils (CCPs), manage these MPAs under provisions of the national fisheries law, with the sanction of the Provincial Ministry of Fisheries, enforcement by the Coastal Police, and support of the NGO Ocean Revolution Mozambique. The CCPs have the first level of enforcement responsibility and administer customary fines from violators. They have the authority to confiscate illegal fishing gear, which is then handed to and destroyed by the Coastal Police. The CCPs have the responsibility of informing and seeking approval for the

implementation of the management plan from the resource users and from all members of the local communities.

**Table 8.** Details on the 12 MPAs created in Inhambane Bay.

| N° | Location | LMMAs Area (in ha) | Year Gazzete | Observation |
|---|---|---|---|---|
| 1 | Guibele | 4.9 | 2017 | North IB |
| 2 | Guidwane | 120 | 2017 | North IB<br>Near Inhambane Island, the greatest gleaning center at the Bay. |
| 3 | Guidzivane | 53.4 | 2017 | North IB<br>With adjacent gleaning areas |
| 4 | Jogó | 118.4 | 2020 | North IB |
| 5 | Thumbine | 7.7 | 2017 | North IB |
| 6 | Chamane | 196.6 | 2020 | South IB |
| 7 | Guindzive | 100.9 | 2017 | South IB |
| 8 | Guilalene | 89.4 | 2017 | South IB |
| 9 | Marragane | 340.3 | 2017 | South IB<br>With adjacent gleaning areas |
| 10 | Maxixe | 45.2 | 2020 | South IB |
| 11 | Ponte Cais | 84.6 | 2020 | South IB |
| 12 | Torotoro | 11.4 | 2020 | South IB |
| | Total area | 1172.8 | | |

The CCPs play a crucial role in raising awareness and engaging communities in resource management. In addition to the norms established by law, communities adopt their own customary practices and rules in the management of invertebrate fisheries. Within Guiduane Island, for example, gleaning is prohibited on Sundays, and the minimum catch size is being disseminated. Within Inhambane, gleaning and bone fish fishing are both equally considered a real fishing activity. Therefore, women are formally called fisherwomen and are included as members of the CCPs. This inclusion facilitates influencing the fisheries sector in gleaning management through the CCPs, allowing the information transmission chain (made of several instruments, campaigns, awareness) to flow faster and more effectively, from the government authorities and NGOs working on the fisheries management of the bay.

One aspect that goes along with ensuring best practices in LMMAs management and conservation are the parallel programs of alternative livelihoods and capacity building. The Ocean Revolution Mozambique (ORM), involved in marine environment management in IB, has been promoting agro-livestock, youth capacity building, and savings and revolving credits, among others. Sensitization is an important component that helped reduce the practice of using drag nets and destructive practices in gleaning within seagrass meadows. The local government, represented by the SDAE (District Services for Economic Activities), which responds for both the fisheries and environment sector at the provincial level, is aware of the LMMAs activities. These LMMAs are rather new, and more assessment may be needed given the existence of the new Maritime Fisheries Regulation (REPMAR). Despite Inhambane Bay having several NGOs, ORM seems to have quite effectively mastered the link between local communities and provincial authorities. Notwithstanding the above-mentioned institutional and governance setup including the engagement of several actors, gleaning activity may need more enforcement; alternative livelihoods may need to be pursued and verified to enable a reduction in the high number of gleaners in the area.

For the western MB (north of Maputo), there is no clear strategy for clam management. However, there is a UNEP regional demo project on seagrass restoration aimed to give recommendations for a tentative management plan for these seagrass meadows. The inclusion of best practices for managing the gleaning activity may be seen as an avenue that can help support the drafting of a seagrass and clam management plan. Actors such as Maputo municipality as well as NGOs and private sectors downstream of the

seagrass value chain are potential key actors in the vision for the seagrass management plan within western MB. Inhaca (eastern MB) seagrass management will also relay on the newly formed (in 2021) community-based organization (named A-TANYI) devoted to the seagrass management of wild stands and restored stands.

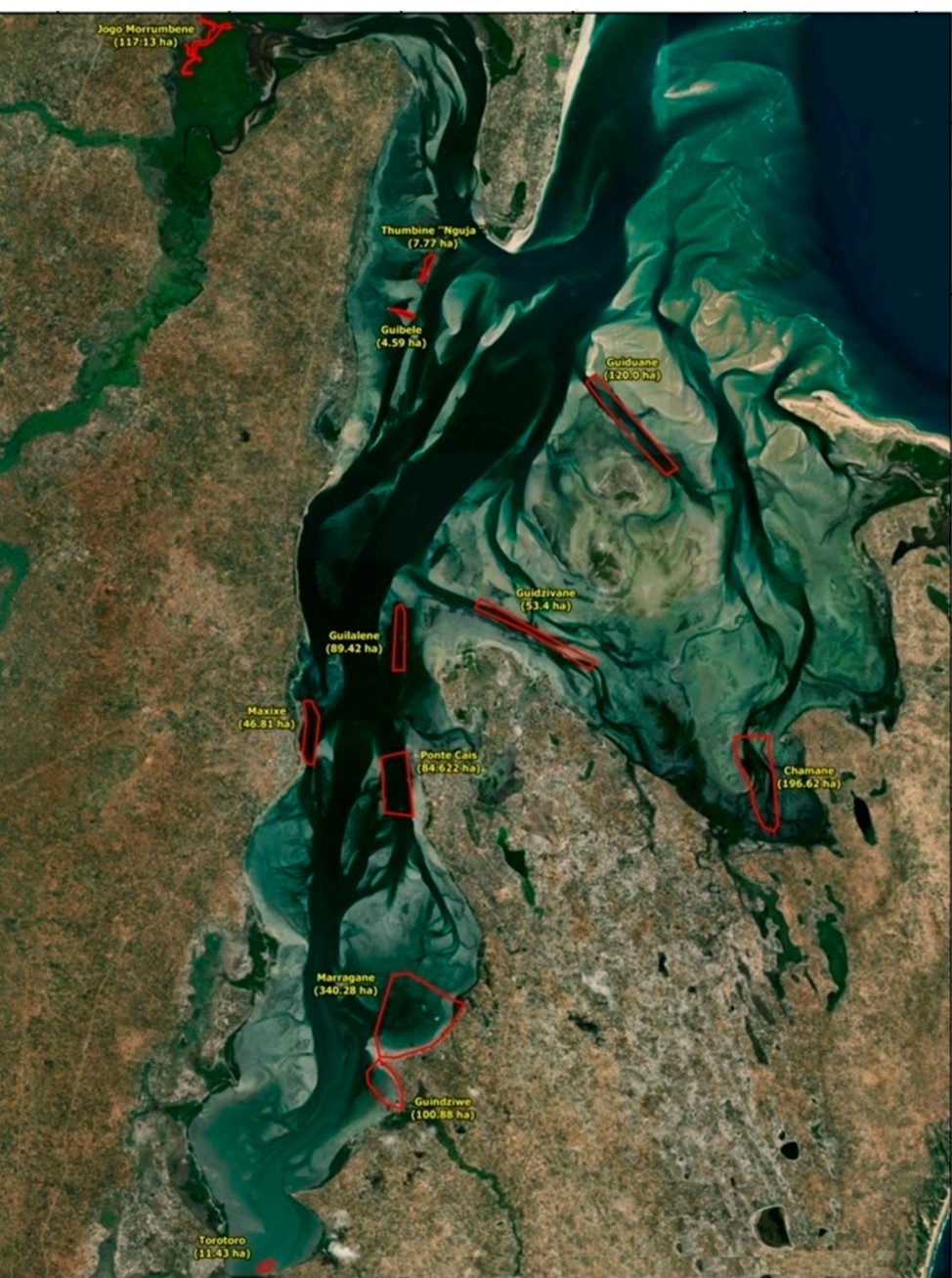

**Figure 5.** Marine areas under community protection (or LMMAs) within Inhambane Bay, outlined in red. Source: Ocean Revolution Mozambique (ORM), 2021.

From a regional point of view, especially following regional seas approach (UNEP, Nairobi Convention applied to the Western Indian Ocean), and global platforms such as SDGs, essential or critical habitats (such as seagrasses, rocky shores, mangroves, coral reefs, and coastal dunes vegetation) need to have well-developed management and monitoring activities. The main items for such documents are: (i) a situational analysis to highlight the status and threats; (ii) mapping of existing entities and key actors; (iii) bringing up guiding pillars and inclusion the vision/mission; (iv) delineating attainable goals; (v) legal

instruments and possible existing community norms; and (vi) a plan of action including indicators, budgets, and timelines.

## 4. Discussion

### 4.1. Resource Use and Depletion

This study highlights the great value of seagrass invertebrate fisheries for the livelihood of coastal communities [22], and the tangible role MPAs can play in supporting the increment of small-scale fisheries [43]. Ref. [39] describes gleaning as an extremely important activity for coastal communities, despite known depletion of resources as documented early within Maputo Bay (5), given the poverty and sometimes full dependence of shallow-water marine resources. Mozambique has over 60% of its people living on the coastal area, being one of the least developed countries yet also highly impacted with issues of climate change, poor management of marine resources, and weak governance systems.

Out of the 89 assessed invertebrate species, 32 were harvested (for IB and Western MB); Inhaca Island reached only 22 common species [6], Madagascar had 34 [44], and Kenya had 158 species [45]. Outside the region, we still found differences comparing, for example, with Indonesia with up to 55 species [46,47] and New Caledonia, which documented 60 invertebrate species in catches [48].

The localized disappearance of community species within intertidal areas in Mozambique has already been reported, with the bivalves Macoma litoralis and Anadara antiquata no longer found due to land reclamation within Maputo bathing areas [37].

A reduction in the quantity and size of invertebrate catches is a problem not only in Mozambique but also in several countries inside and outside the WIO; looking at studies carried out on seagrasses, the high number of gleaners (average numbers per day are 80 for MB and 40 for IB-Guidwane) leads to overfishing, which is one of the most important factors mentioned as being related to these declines [44,46].

The degradation of seagrasses leads to the depreciation of resources that are dependent on them; therefore, seagrass restoration as a way to restore fisheries is a scientifically viable option [37]. This study, together with the previous studies carried out in the study areas, paved the way for seagrass restoration [35,49] and documentation towards seagrass management [50]. The authors of [50], who compared both coastal and inland areas in west and central African fisheries, highlighted the modeled value chains to underpin social issues, actors that can support the desired sustainability of fisheries, and an analogy for the case of Guidwane within IB underpinning several functional social, economic, and environmental issues.

There is a science challenge on issues of learning and adapting to similar experiences of the overexploitation of seagrass-meadow-associated resources such as clam catches. Potential options for aquaculture need to be looked at and tested. Such initiative can be tested with the involvement of a research/academic institution. Socio-anthropological research may also document and support alternatives for sustainable clam harvesting and related community aquaculture schemes such as the left-on-field clams, to grow for later collection (anecdotal accounts from some community members at Bairro dos Pescadores).

### 4.2. Onset of LMMA's and Seagrass Management

Both MB and IB suffer from a continued reduction in resources, as confirmed in previous studies from Inhaca [5] and documented elsewhere [51]. Inhambane proclamation of LMMAs, although each being small areas of LMMA, signals the revamping of positive community intervention in sustainability and the conservation of critical habitats and shallow-water resources. However, the devised LMMAs need to be coupled with the active intervention of actors such as the government and civil society [22]. Inhaca Island within Maputo Bay is part of the wider protection areas, recently integrated in the Partial Marine Reserve of Ponta do Ouro (RMP-PO) [35], but the depletion of resources may still be relentlessly exacerbated by an extreme reduction in tourism due to COVID-19. According to [14], the easy and free access to seagrass meadows favors the overexploitation

of resources. The authors of [4] showed that the direct exploration of invertebrates can alter and reduce the biomass and density of resources associated with seagrasses, even if only at a subsistence level. LMMAs are a non-formal protection of marine areas [17], basically being enforced by the communities themselves; therefore, these areas are not formal MPAs. General information regarding regular governance, funding [52], but also resource assessment and levels of community wellbeing and satisfaction need to be documented in order to evaluate the effectiveness of these LMMAs [26]. Mechanisms for seagrass resource management need to be widely discussed and disseminated. Gleaner's discontent within Guiduane Island (part of IB) is because they have to compete with harvesters from other areas within the same bay. Islanders depend exclusively on fishing; the only income activity for most women is the collection and sale of shellfish, and they claim to know all the measures to be taken for sustainable fishing better than most. The large number of non-resident gleaners uses destructive gleaning methods, and the catching of juvenile fish therefore contributes to fishery decline.

Small-scale seagrass fishery resources are immense given their socio-anthropological, physical, and also seasonal boundaries linked to communities [53]; however, they are vulnerable due to the reported resource depletion [5]. Across the globe, experiences of seagrass management are generally incorporated into the wider coastal, shallow-water systems [54,55] with a focus within the tropics, for mangroves forests [56] or major impacts such as erosion [57]. NW Maputo Bay seagrass meadows are basically overseen by local authorities with little intervention of the municipality. A seagrass management plan harmonized with, for example, regional seas discussion under UNEP can be a way forward, and such a management plan for MB may include:

1. The development of community-based management (CBM) for invertebrate fisheries—a bottom-up approach is needed to secure the sustainability of this fishery;
2. Conservation area authorities, municipalities, NGOs, and research/academic institutions need to guarantee the appropriate implementation and wider best practices, as well as social adaptation to guarantee long-term sustainability and a change in the culture of clam collection;
3. Wider awareness/sensitization on seagrass meadows and their fisheries;
4. Promote discussion on resource extraction, gear used, and possible discussion on quotas, value chains, and community development.

In the NW Maputo Bay, the extraction of invertebrates is acute, and this part of the bay should be the main target for the Maputo Bay seagrass management plan, and options for aquaculture of clams could be a priority; test experimentation and socio-anthropological as well as stakeholder analysis are needed.

## 5. Conclusions

This study on invertebrate gleaning within seagrass meadows brought more evidence about the diversity of species and community, the fishing techniques and gear, the economic value chain, and sociocultural dimension of the gleaning activity. The collection of molluscs in the BM and BI intertidal zones is an activity with a female tradition, characterized by people of low economic status, exclusively by coastal communities and destined to subsistence, but they also contribute significantly to the household's income. The search for greater community needs requires the collection of large quantities of molluscs, from around 1.5 metric tons to 1.9 metric tons/year per collector (in western MB and IB, respectively), overexploitation that put into question the production capacity of seagrass ecosystems. Vast seagrass areas are disturbed, and the seagrass area uprooted is 2.74 ha/year in the western MB and 8.80 ha/year in IB. Coastal areas across the tropics are also experiencing population growth.

Considering the estimates of the quantities of invertebrates that circulate in market networks, it seems relevant to include these artisanal fisheries in the wider official statistics to inform ongoing discussion on oceans governance in Mozambique and the Western Indian Ocean, supporting raising awareness and guiding protection of seagrass meadows

with wider involvement of communities and society. The ongoing experience of LMMAs paves the way for an informed use of local marine resources; however, from other tropical experiences, Mozambique LMMAs initiatives require the continued engagement of actors in a conservation network fashion, as well as financial backup to sustain sensitization, discussion, and more important sustainability actions and alternative livelihoods. Issues of poverty alleviation, gender empowerment, and formal education, especially for girls, should support increasing wellbeing of the coastal communities.

The main recommendation is to empower LMMAs to be more interventional in the management of ecosystem services. This can be achieved by establishing a region LMMAs network, a forum where different actors may converge in mainstreaming necessary solutions for conservation, livelihood sustainability, and population wellbeing. There is a need for a continuous understanding and support of the value chains so that the most advantaged find meaningful gains as resource extraction becomes more sustainable and favours gender. Following a global approach and platforms, more research is needed, especially on the carrying capacity of gleaners and productivity of shallow seagrasses meadows to address a wider strategy of seagrass management in Mozambique.

**Author Contributions:** Conceptualization, S.B. and S.C.-N.; methodology, S.C.-N. and S.B.; formal analysis, A.C. and S.C.-N.; fieldwork, S.C.-N., A.C., M.E.M. and A.d.S.C.; data curation, A.C. and S.C.-N.; writing—review and editing, M.E.M., A.G. and S.B.; supervision, S.B.; project administration and funding, S.B. and A.d.S.C. All authors have read and agreed to the published version of the manuscript.

**Funding:** This study was funded by WIOSAP/UNEP (Grant Number: GEF Project ID: 4940; SSFA/2019/2480), MASMA "seagrass protect"/WIOMSA (Grant n°: MASMA/OP/2018/02) and AKDN—Fundação Aga Khan and FCT—Fundação para a Ciência e a Tecnologia, IP for funding the COBIO-NET project. APC was funded by WIOSAP/UNEP (Grant Number: GEF Project ID: 4940; SSFA/2019/2480), MASMA "seagrass protect"/WIOMSA (Grant n°: MASMA/OP/2018/02).

**Institutional Review Board Statement:** Not applicable.

**Informed Consent Statement:** All interviewees, received a written statement explaining the purpose of the study. The informed consent statement highlighted that the participation in the study was voluntary and there were no known risks to participation beyond those encountered in everyday life. It also underlined that participants could decline altogether or leave blank any questions they did not wish to answer and their responses would remain confidential and anonymous. Only those who provided written or oral consent were interviewed.

**Data Availability Statement:** Data from this research is kept under lock key at the Department of Biological Sciences Research Database. It will be available for re-use by researchers in future studies.

**Acknowledgments:** We thank the support from IPSN (Indo-Pacific Seagrass Network). G. Cassamo georeferenced Inhambane LMMAs. J. Campira created the study area map. Part of data for this study was collected within the scope of graduation of both SCN and AC. We extend our vote of thanks to the anonymous reviewers.

**Conflicts of Interest:** The authors declare no conflict of interest, all parts involved including the sponsors support the publication of results. The study was solely designed, implemented and written by the authors.

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
