# Peer review of "Seagrass Invertebrate Fisheries, Their Value Chains and the Role of LMMAs in Sustainability of the Coastal Communities—Case of Southern Mozambique"

_diversity, doi:10.3390/d14030170_

Round 1

Reviewer 1 Report

This review is for the manuscript titled “Seagrass invertebrate fisheries, their value chains and the role of LMMAs in sustainability of the coastal communities – case of southern Mozambique” as submitted to the journal Diversity.

The manuscript describes invertebrate gleaning activity from two large bays, including information about the physical environment, regulatory policy framework, invertebrates present and gleaned, and processing and marketing of invertebrates.  The manuscript advocates for conservation management of these and similar bays.

I like the manuscript and the subject.  My recommendations below are toward reorganizing the manuscript into a better format.  I did not make detailed suggestions regarding the English, but overall it is well written for publication.

This is a descriptive study with focus on conservation policy.  I defer to the Editor whether the material is a good fit for this journal, and whether the length of the manuscript is good or should be reduced.  I would be willing to review again after comments from Reviewers and Editors are addressed.  Thank you for the opportunity to review.

P1L41: Species names are italicized elsewhere in the document; check the guide for authors for this journal and be consistent throughout.

P5L165: Spring tide was confusing here, as I initially thought it meant gleaning only occurred in spring rather than throughout the year.  Would it be accurate to use “full moon tide” rather than “spring tide”?

P12 Table 7: Do you have ideas why one location was so much more responsive regarding mitigation measures?  Add a little to the Discussion section on this.

P12-16: These 12 paragraphs and Table 8 and Figure 4 should be moved to the beginning of the Materials and Methods into a new section 2.2 titled Policy Framework.

P16 First paragraph in 4.1: This paragraph should be moved to the second-to-last paragraph of the Introduction.

P18 This section is not finished.  The last five paragraphs are not complete.

Author Response

The authors would like to thank the Editor and the Reviewers for their detailed and insightful comments, and we hope that these improvements have adequately addressed all the concerns.

We look forward to hearing from you at your earliest convenience.

Sádia Chitará Nhandimo, on behalf of all the authors.

Reviewer 2 Report

The paper by Chitara-Nhandimo et al entitled "Seagrass Invertebrate fisheries, their value chains and the role of LMMAs in sustainability of the coastal communities – case of southern Mozambique" is dealing with the importance of gleaning in local population and how preservation resources through local/national laws and MPAs are in dear need. Although the article is very interesting, the paper suffer several flaws that need be addressed by the authors before publication. A thorough reading and correction by someone fluent in English will definitively help. Please check also for typos, i.e. missing spaces, , instead of . for numbers, sentences way to long, sentences missing verb.
The authors needs also to better take advantage of the data collected. I checked all the scientific names of the species and over 50% were easier wrong or misspelled, two species are not valid. They mention quantitative approach but no values regarding total catches/sales are presented.

A better presentation of the sampling areas and collection tools are also needed.

I made some comments on the PDF file, which is attached to my review.

Author Response

The authors would like to thank the Editor and the Reviewers for their detailed and insightful comments, and we hope that these improvements have adequately addressed all the concerns.

We look forward to hearing from you at your earliest convenience.

Sádia Chitará Nhandimo, on behalf of all the authors

Reviewer 3 Report

Please see the PDF attachment.

Author Response

(The authors gave the same response as above.)

Round 2

Reviewer 1 Report

Thank you for your contribution to this subject.